# The Ecological Water Demand of *Schizothorax* in Tibet Based on Habitat Area and Connectivity

**DOI:** 10.3390/ijerph16173045

**Published:** 2019-08-22

**Authors:** Zili Zhou, Yun Deng, Yong Li, Ruidong An

**Affiliations:** State Key Laboratory of Hydraulics and Mountain River Engineering, Sichuan University, Chengdu 610065, China

**Keywords:** connectivity, ecological flow, habitat, *Schizothorax*, spawning period

## Abstract

Water resource regulation is convenient for humans, but also changes river hydrology and affects aquatic ecosystems. This study combined a field investigation and two-dimensional hydrodynamic model (MIKE21) to simulate the hydrodynamic distribution from 1 March to 30 April of 2008–2013 and establish the HDI (habitat depth suitability index) and HVI (habitat velocity suitability index) based on static hydraulic conditions at typical points. Additionally, by using MIKE21 to simulate the hydraulic state in the study area under 20 flow conditions from 530–1060 m^3^/s, and combining these states with the HCI (habitat cover type suitability index), HDI, and HVI, we simulated the WUA (weighted usable area) and habitat connectivity under different runoff regulation scenarios to study the water requirements of *Schizothorax* during the spawning period in the Yanni wetland. The results showed the following: (1) the suitable cover type was cobble and rock substrate, with nearby sandy land; furthermore, the suitable water depth was 0.5–1.5 m, and the suitable velocity was 0.1–0.9 m/s. (2) Using the proximity index to analyse the connectivity of suitable habitats, the range of ecological discharge determined by the WUA and connectivity was 424–1060 m/s. (3) Habitat quality was divided into three levels to detail the flow demand further. When the flow was 424–530 m^3^/s or 848–1060 m^3^/s, the WUA and connectivity generally met the requirements under natural conditions. When the flow was 530–636 m^3^/s or 742–848 m^3^/s, the WUA and connectivity were in a good state. When the flow was 636–742 m^3^/s, the WUA and connectivity were in the best state. This study complements existing research on the suitability of *Schizothorax* habitat in Tibet, and introduces the connectivity index to enrich the method for calculating ecological water demand, providing a reference for resource regulation and the protection of aquatic organisms.

## 1. Introduction

The intensity of the exploitation and utilisation of water resources increases with continuous economic development [1]. The runoff regulation alleviates the frequency of flood disasters and the uneven distribution of water resources but also has a strong negative impact on the ecological environment. Reservoirs change downstream runoff processes, which directly or indirectly affect habitat quality and change the structure, composition, and distribution of the biological community [2,3]. A lack of ecological discharge causes the lower reaches to dry up, reduce the number of species, and deteriorate the ecological environment. For example, construction of the Aswan Dam on the Nile River in Africa accelerated coastal erosion and salinisation [4], and different discharge rates of the Glen Canyon Dam in the United States affected the spawning and hatching rate of rainbow trout [5].

The Yarlung Zangbo River originates from the Jemayangzong Glacier in the Himalayas in the southwest of Tibet, with high altitude, cold temperatures, strong radiation, and other extremes, resulting in a unique and fragile ecosystem [6]. The annual average runoff of this river in China is 166 billion m^3^, with an uneven distribution throughout the year. As water resource regulation is imperative, it is urgent to study the ecological water demand in Tibet to ease the conflict between the development and ecological sustainability.

Fish represent the apex community in many aquatic ecosystems and are an important aquatic biological resource [7]. Dynamic changes in the number and population structure of fish can comprehensively and effectively reflect the overall changes in aquatic organisms and water quality status [8]. Thus, studies commonly use fish as a proxy to analyse aquatic ecological water demands. Tibetan *Schizothorax* is of great scientific value for studying topographic changes, evolution, and extreme environmental adaptation mechanisms on the Tibetan Plateau [9]. *Schizothorax* is also one of the main economic fishes in Tibet [10], with a slow growth rate, poor adaptability to environmental changes, and late sexual maturity (usually 3–6 years) [11], and is mainly distributed in the Yarlung Zangbo River. The spawning period of *Schizothorax* is 3–4 months, with a peak in March [10,12].

Currently, based on a global review, environmental flow methodologies could be classified as follows [13]: hydrological [14,15], hydraulic rating [16,17], habitat simulation [18,19], and holistic [20]. The habitat simulation method, including IFIM (Instream Flow Incremental Methodology) and CASIMIR (Computer-Aided Simulation Model for Instream Flow Requirements In Diverted Stream), associates the physical habitat (e.g., depth and velocity) with biological response indicators (e.g., oviposition and fertilization). The resultant outputs of the habitat simulation models, mainly including the PHABSIM (physical habit simulation model) [21], the RHYHABSIM (river hydraulics and habitat simulation system) [22], and River-2D [23], are often depicted as effective habitat time series and duration curves and used to quantify EFRs (environmental flow requirements) and evaluate alternative flow regulation scenarios [13].

Hydraulic characteristics are the main factors defining fish habitat and the biological composition [24]. Choosing suitable hydraulic parameters as indicators to characterise spawning habitat requirements is important for ensuring that the habitat quality simulated by environmental flow methodologies is accurate under different flow conditions. Scholars in China and elsewhere have used various hydraulic parameters to describe the characteristics of fish spawning grounds, including water depth [25], velocity [25,26], the Froude number [25,27], the kinetic energy gradient [27,28], and vortex intensity [29]. Appropriate water depth can create space for fish to move within and provide a suitable incubation environment for viscous fish eggs, and appropriate water velocity can stimulate fish to lay eggs and affects the gonadal development of fish, which requires sufficient dissolved oxygen [30]. Therefore, we used water depth and velocity to represent hydraulic requirements and to analyse habitat suitability in this study.

The existing habitat simulation method considers the suitable area demand without considering suitable habitat size or spatial connectivity. The movements of individuals, materials, nutrients, energy, or disturbances through a landscape involve more than the boundary configuration, permeability, context, and patches arrayed in the mosaic. The probability that an organism or an ion at one location in a landscape will move to some other location is a function of the combination of patch types and boundaries that separate those locations [31]. Composite metrics, such as the WUA (weighted usable area), may not sufficiently define the conditions of the river environment. The spatial distribution of aquatic habitat and size of habitat patch are also important [32]. The relationship among habitat patches could be quantified using connectivity metrics based on the size, arrangement and hydrographic distances of habitats to complement traditional habitat measures [33].

Because of the fragile habitat and urgent need for ecological water demand research in Tibet, in this study, we determined the HSI (habitat suitability index) of the substrate and hydraulic conditions during the spawning period based on a field investigation, literature investigation, and numerical simulation. We calculated the ecological water demand using the habitat simulation method. Additionally, the connectivity calculation method from landscape ecology was employed to analyse the quality of suitable habitat and enrich the modelling of ecological water demand.

## 2. Materials and Methods

### 2.1. Study Area

The Yanni wetland is located at the confluence of the Yarlung Zangbo River and the Niyang River (Figure 1), and contains diverse habitats and rich species; the mainstream part of the wetland is an important area for *Schizothorax* spawning and hatching [34]. The mainstream section of the Yanni wetland (from Channa to Nuxia) was used as the study area. In the section, the length of the reach was approximately 16 km, the width was 1450–3350 m, the perennial water surface width of the river was 427–1282 m, and the maximum water depth was 5.6–16.1 m. The river is of a wide valley type, with developed terraces on both sides and beaches in the middle.

The river channel is dominated by bifurcated-compound channels and is developed with terraces on both sides (Figure 2). During the period of normal water level, there are two main steps composed of cobble and rock substrates. The river channel develops into sandy land, grassland, shrubland, and farmland on both sides. In the non-flood season, the water is mainly distributed in the main channel, and in the flood season, the flow significantly increases, flooding the beaches.

### 2.2. Data

According to survey results, the main fish species in the reach are *Schizothorax* (94.2% of the catch), including *Schizothorax oconnori Lloyd* (EN (endangered, [35]), 65.8%), *Schizothorax macropogon Regan* (EN, 26.7%), and *Schizothorax watoni Regan* (EN, 2.1%). The length of the fish ranged from 123 to 504 mm, and the spawning period was from March to April, with a peak in March.

Based on the daily flow data from the Nuxia hydrological station from 2000–2013, Figure 3 shows the flow frequency curves in March and April. The average flow during the spawning period was 530 m^3^/s (expressed below as “Q”). The flow in March was stable (600–400 m^3^/s), and the 90% guaranteed frequency flow was used as the lower limit of flow in the spawning period (420 m^3^/s, 0.8 Q). In April, the flow and its amplitude increased due to the snowmelt and rainfall.

### 2.3. Methods

Figure 4 shows the flowchart of the research. Based on the topographic and hydrological data, a field investigation, literature research, and satellite processing, we determined the habitat suitability curve, WUA, and habitat connectivity during the spawning period of *Schizothorax*; and finally determined the ecological water demand during this period.

#### 2.3.1. Hydrodynamic Model

For rivers, where the horizontal scale is much larger than the vertical scale, changes in hydraulic parameters such as water depth and velocity in the vertical direction were much smaller than those in the horizontal direction. Two-dimensional plane models could be used to simulate the hydraulic conditions of rivers by homogenizing the three-dimensional governing equations along with the water depth. Among these models, MIKE21, developed by the DHI (Danish Hydraulic Institute), is widely used worldwide [36,37,38,39].

We selected the MIKE21 hydrodynamic model to simulate the distribution of water depth and velocity in the region based on wide cross-sections of the study area. The results were used to determine the suitable curve of hydraulic habitat during the spawning period of *Schizothorax* and to provide hydrodynamic data for the calculation of the WUA.

The MIKE21 model is based on three Navier–Stokes equations uniformly distributed by incompressible and Reynold’s values, and the model is subject to the Boussinesq assumption and hydrostatic pressure assumption:
(a)Continuity equation:(1)$$\frac{\partial h}{\partial t}+\frac{\partial hu}{\partial x}+\frac{\partial hv}{\partial y}=hS$$(b)Momentum equation:(2)$$\frac{\partial h\bar{u}}{\partial t}+\frac{\partial h{\bar{u}}^{2}}{\partial x}+\frac{\partial h\bar{uv}}{\partial y}=f\bar{v}h-gh\frac{\partial \eta }{\partial x}-\frac{h}{\rho_{0}}\frac{\partial p_{a}}{\partial x}-\frac{gh^{2}}{2\rho_{0}}\frac{\partial \rho }{\partial x}+\frac{\tau_{sx}}{\rho_{0}}-\frac{\tau_{bx}}{\rho_{0}}\;-\frac{1}{\rho_{0}}\left(\frac{\partial S_{xx}}{\partial x}+\frac{\partial S_{xy}}{\partial y}\right)+\;\frac{\partial }{\partial x}\left(hT_{xx}\right)+\frac{\partial }{\partial y}\left(hT_{xy}\right)+hu_{s}S,$$
(3)$$\frac{\partial h\bar{v}}{\partial t}+\frac{\partial h\bar{uv}}{\partial x}+\frac{\partial h{\bar{v}}^{2}}{\partial y}=-f\bar{u}h-gh\frac{\partial \eta }{\partial x}-\frac{h}{\rho_{0}}\frac{\partial p_{a}}{\partial y}-\frac{gh^{2}}{2\rho_{0}}\frac{\partial \rho }{\partial y}+\frac{\tau_{sy}}{\rho_{0}}-\frac{\tau_{by}}{\rho_{0}}\;-\frac{1}{\rho_{0}}\left(\frac{\partial S_{yx}}{\partial x}+\frac{\partial S_{yy}}{\partial y}\right)+\;\frac{\partial }{\partial x}\left(hT_{xy}\right)+\frac{\partial }{\partial y}\left(hT_{yy}\right)+hv_{s}S,$$
where *η* is the water level; *d* is the static water depth; *h* = *η* − *d* is the total water depth; *u* and *v* are the velocity components (m/s); f=2ωsinφ is the Coriolis force, where ω is the rotational angular velocity of the earth and φ is the local latitude; g is the gravity (m^2^/s); *ρ* is the water density; *S_xx_*, *S_xy_*, and *S_yy_* are the respective radiation stress components; *S* is the source; *us* and *vs* are the velocity of the source, respectively; and *T_ij_* is the horizontal viscous stress term.

#### 2.3.2. Habitat Model

(1) Habitat Suitability Index:

(a) Channel suitability index

Only by adhering to cobbles and other substrates can *Schizothorax* fish eggs hatch successfully and undergo short-distance reproductive migration [40]. According to previous studies, the spawning area of *Schizothorax* in Tibet is generally located in places where the riverbed gravel is relatively thick and is covered by pebbles or stones with small amounts of silt [10,12]. Young fish forage in slow flows on sediments, usually in areas with higher water temperatures, such as a migratory bays, wide valleys, and sinks [34].

Mark Hampton divided the cover type criteria of channels into cobble, boulder, and wood [41]. In this study, considering that increased flow may cover the floodplain, and according to the field investigation and satellite image interpretation technology, we confirmed the cover type throughout the entire riverway within the research area; the cover types included rock, cobble, silt, grassland, shrub, agricultural land, and forest.

(b) The depth and velocity suitability indices

Existing studies suggest that spawning areas contain interconnected areas of rapid and slow flows [42,43], with water depth of approximately 0.3–1.5 m [10,12]. These results do not meet the accuracy requirements of habitat simulation but provide a reference for the establishment of a hydraulic suitability index.

With a developed floodplain, rich hydraulic conditions, and scattered spawning areas, in the present study area, we used typical points to construct the HSI of hydraulic conditions, which is similar to the real requirements but also requires higher accuracy in the selection of monitoring points. Considering the results of existing hydraulic habitat research, the investigation of spawning areas in the study area, and the topographic data, the *Schizothorax* spawning areas are scattered throughout the study area, and most of the sites are distributed near the edge of the beach and in the middle of the beach at the junction of fast and slow currents (i.e., at the second step area in Figure 2). In the selection of typical points, we considered the following: (1) the fast flow rate in the main channel provides simulation signals for spawning and hatching. (2) The flat terrain and slow flow rate on the beach facilitate predation of young fish. (3) The water mixing at the junction of the main channel and beach is strong, which is conducive to the fertilisation of fish eggs. Therefore, we chose this junction as the typical point (shown in Figure 5). On this basis, we selected points that met the requirements for determining typical points among all cross-sectional topographic data in the study area and formed a set of typical points, the spatial distribution of which is shown in Figure 5. Subsequently, MIKE21 was used to simulate the water depth and flow velocity of the typical points from 1 March to 30 April from 2009–2013. After eliminating the hydraulic calculation results (from existing research on hydraulic habitat demand) whose water depth was not between 0.3 and 1.5 m, the number of samples at the different water depths and velocity ranges was counted, and the statistical results were normalised to establish the hydraulic suitability index during the spawning period of *Schizothorax*.

(2) Suitable Area

By combining the distribution of water depth and velocity simulated by MIKE21 with confirmed cover types within the study area, and according to the HSI of the cover type, depth, and velocity, we determined the CSI (comprehensive habitat suitability index) and simulated the UA (usable area) and the WUA under different runoff regulation scenarios:(4)$$CSI_{i}=\sqrt[{3}]{HCI_{i}\times HDI_{i}\times HVI_{i}}$$
(5)$$\mathrm{UA}=\overset{k}{\underset{j=1}{\sum }}A_{j}$$
(6)$$\mathrm{WUA}=\overset{n}{\underset{i=1}{\sum }}\left(A_{i}\times CSI_{i}\right)$$
where *HCI_i_*, *HDI_i_*, and *HVI_i_* are the cover type suitability, depth suitability, and velocity suitability of the *i*th grid, respectively; *k* is the number of grids where *CSI_j_* > 0; *A_j_* is the area of the *j*th grid where *CSI_j_* > 0; *k* is the number of grids; and *A_i_* is the area of the *i*th grid.

Table 1 shows five habitat suitability levels, which are divided based on CSI values. As suitability increases, ecological significance increases. The total habitat was used to analyse the variation in the total WUA and connectivity and to determine the ecological flow; however, different suitable habitats were used to enrich and refine the ecological flow demand.

#### 2.3.3. Connectivity Model

In this study, we used the PROX (proximity index) to analyse habitat connectivity. PROX describes the spatial relationships between different habitat patches [33] and distinguishes between sparse distributions of small patches and clustered distributions of large patches [44]. In a landscape, patches refer to the spatial units that are different from the surrounding environment in appearance or nature and have certain internal homogeneity, which affects biomass, species composition, and diversity. Patches of the same nature are combined into a class. In this study, patches refer to the regions with a CSI >0 and are divided into five classes according to the range of CSI value (Table 1). The patch connectivity of each spawning habitat was determined by PROX*_ij_*, and each PROX*_ij_* value was weighted by the area to calculate the PROX*_i_* of class *i*:(7)$${\mathrm{PROX}}_{ij}=\overset{m}{\underset{s=1}{\sum }}\frac{a_{ijs}}{h_{ijs}^{2}}$$
(8)$${\mathrm{PROX}}_{i}=\overset{n}{\underset{j=1}{\sum }}\frac{a_{ij}}{A_{i}}\times {\mathrm{PROX}}_{ij}$$
where PROX*_ij_* is the proximity index of the *j*th patch in class *i*; *m* is the number of patches within the specified neighbourhood (*r*, *m*) of patch *ij*; *a_ijs_* is the area (m^2^) of patches *ijs* within the specified neighbourhood (*r*) of patch *ij*; *hijs* is the distance (m) between the patch *ij* and patch *ijs*; PROX*_i_* is the proximity index of class *i*; n is the number of patches in class *i*; *aij* is the area (m^2^) of patches *ij*; and *A_i_* is the area (m^2^) of class *i*.

The specified neighbourhood (*r*, *m*) must be identified before the PROX calculation. As the spatial distribution of aquatic habitat quality and size is an important factor affecting the quality of aquatic habitat in a 0.01 m resolution macrohabitat [32] and the channel in the study area is wide, we set the specified neighbourhood to 100 m (*r* = 100).

PROX ≥0 is dimensionless and is used as a comparison index. PROX = 0 indicates that there are no other patches of the same type around the patch in the specified neighbourhood. PROX increases when there are more homogeneous patches in the neighbourhood and when there is more compact distribution.

## 3. Results

### 3.1. Hydrodynamic Model

We used the data of the water boundary line measured in the study area in May 2013 to carry out parameter determination for the MIKE21 hydrodynamic model. Figure 6 shows the verification results of the model, where points 1 to 11 are water edge extraction points determined by field investigation, which are different from the typical points in the habitat model. As the water level error is 0–0.1 m, the model can well represent the water dynamics in the study area. 

Figure 7 shows the distribution of water depth in the study area under the natural flow rate (1 Q): the water depth is distributed in the main channel, sub-channel, etc., while the velocity is distributed only in the main channel, as the sub-channel is mostly in a static state under the influence of the floodplain. Compared with the natural river network structure (Figure 5), the simulation results in Figure 7 were similar.

### 3.2. Habitat Model

#### 3.2.1. Habitat Suitability Index

(1) Channel suitability index:

Controlled by bifurcated compound channels and terraces on both sides, the cross-section along the elevation in the channel could be divided into the first step, the second step, and land (Figure 2). The first step in the channel mainly consists of rocks, and the cover type does not satisfy the conditions for hatching; thus, the HCI is 0. The second step mainly consists of cobble and develops into sand towards the banks, the cover type is suitable for spawning and hatching; thus, the HCI is 1.

According to satellite data, the cover types of land above the water surface are sandy land, grassland, shrubland, and farmland, none of which meet the spawning requirement; thus, the HCI is 0.

(2) Hydraulic suitability index:

Based on the statistics of the typical points’ hydraulic, simulated by MIKE21, from March to April of 2009–2013, and normalisation processing, we obtained the HDI and HVI (Figure 8). The suitable depth range was 0.5–1.5 m, in which at a depth of 0.6–1.0 m, the HDI >0.6, indicating good suitability, and at 0.7–0.8 m, the HDI >0.8, indicating the best suitability for spawning. The suitable velocity range was 0.1–0.9 m/s, in which at a velocity of 0.3–0.7 m, the HVI >0.6, indicating good suitability, and at a velocity of 0.5–0.6 m, the HVI >0.8, indicating the best suitability for spawning.

#### 3.2.2. Suitable Area

MIKE21 was used to calculate the distribution of the water depth and velocity under different runoff scenarios in the study region. Figure 9 shows the cover type, depth, velocity suitability, and CSI distributions during the spawning period under natural conditions. (a) The suitable cover type is mainly distributed in the second step along the river, with good continuity. (b) Under natural conditions, the distribution of the suitable depth areas is relatively dispersed because of the widely distributed floodplain and bifurcated river channels. (c) Under the influence of the bank-up, the water flow in shoals and grooves was mostly in a static state, and suitable velocities were obviously concentrated in the main channel. (d) Considering the suitability of the cover type, water depth, and flow velocity, the distribution of spawning areas suitable for *Schizothorax* decreased significantly, which was mainly affected by the cover type and water depth requirements.

The total suitability of cover type (HCI_total_), depth (HDI_total_), and velocity (HVI_total_) and comprehensive suitability (CSI_total_) in the statistical region were used to analyse the causes of suitability change in the spawning area of *Schizothorax* under different runoff conditions, as shown in Figure 10.

HCI_total_: when the flow is 0.1–0.5 Q, the water is mainly distributed in the first step of the main channel, where the cover type suitability is 0. When the value increases from 0.6–0.8 Q as the water level rises, the submerged area expands to the second step and HCI_total_ increases. When the value is 0.9 Q, the water level submerges the channel, reaching the maximum HCI_total_. When the value continues rising, and the water level covers the banks, of which the main substrata are sandy land and grassland, which are not suitable for *Schizothorax* fish reproduction, HCI_total_ remains unchanged.

HDI_total_: when the flow is 0.1–0.5 Q, the water depth is suitable within the first step of the main channel. When the value increases from 0.6–0.8 Q, as the water level rises, the depth of the first step is too high, the depth of the second step is shallower than that of the main channel, and the HDI_total_ decreases slightly. When the value increases from 0.9–1.3 Q, the water level rises slowly because of the wide beach, while HDI_total_ is almost unchanged. When the value is higher than 1.4 Q, the water in the channel is too deep, and the suitable area is transferred from the channel to the beaches and grooves, which have better depth suitability.

HVI_total_: when the flow increases to 0.9 Q, the HVI_total_ increases as the velocity and the cross-section area increase. Then, with the increase in flow, the water spreads to the beach, and the velocity suitability of the main channel increases slowly, while the beach flow velocity is slow and the suitability is poor; thus, the HVI_total_ increases slowly.

CSI_total_: when the flow is 0.1–0.5 Q, CSI_total_ is 0 and is mainly affected by HCI_total_. When the value is 0.6–0.8 Q, HCI_total_ and HVI_total_ increase, while HDI_total_ decreases slightly and CSI_total_ rises slowly. When the value is 0.9–1.3 Q, HCI_total_ maintains the maximum value, and HVI_total_ increases slowly, while HDI_total_ is almost unchanged. The overlapping between the categories is intensified, and CSI_total_ increases significantly and reaches a maximum at 1.3 Q. When the value is 1.4–2.0 Q, the suitable depth area develops towards the beach, and the overlaps between the three suitability categories decrease, causing a reduction in CSI_total_.

WUA*_i_* is controlled by the *i*th grid area and CSI. In this model, the total WUA is significantly correlated with CSI_total_, as the area of each grid is relatively uniform (Figure 11, R^2^ = 0.9999).

Figure 12 shows the variation in WUA, WUA_(__0, 0.3)_, WUA_(0.3, 0.6)_, WUA_(0.6, 0.8)_, and WUA_(0.8, 1.0)_ with the flow change. At 0.1–0.5 Q, CSI_total_ = 0, and WUA = 0; thus, there is no suitable spawning area for *Schizothorax*. At 0.6–0.9 Q, the WUA increases as CSI_total_ increases slowly. At 0.9–1.1 Q, WUA increases significantly. At 1.2–1.3 Q, WUA maintains a high value and WUA_max_ = 258 × 10^4^ m^2^ at 1.3 Q. At 1.4–2.0 Q, the WUA decreases as the CSI_total_ is reduced.

The WUA values corresponding to different suitability ranges were calculated, and WUA_(0.8, 1.0)_ changed dramatically with the flow. WUA_(0.8, 1.0)_ significantly increased from 0.2–14.5 × 10^4^ m^2^ as the flow increased from 0.6–1.1 Q. Then, the value remained high from 14–19.2 × 10^4^ m^2^ from 1.1–1.4 Q and was reduced to 0.1 × 10^4^ m^2^ when the flow reached 2.0 Q. WUA_(0.6, 0.8)_ significantly increased from 0.4–9.5 × 10^4^ m^2^ as the flow increased from 0.6–1.1 Q; it maintained a value from 9.2–9.9 × 10^4^ m^2^ from 1.1–1.4 Q and then fell to 0.3 × 10^4^ m^2^ when the flow reached 2.0 Q. WUA_(0.3, 0.6)_ increased from 0.4–14.3 × 10^4^ m^2^ as the flow increased from 0.6–1.1 Q, and it slightly changed (+0.8, 1) from 1.1–1.7 Q and then decreased to 7.4 × 10^4^ m^2^ at 2.0 Q. WUA_(__0, 0.3)_ maintained a lower value or even a value of 0 under all flow conditions.

According to the change in WUA, the ecological water demand of *Schizothorax* during the spawning period was determined: (1) the spawning peak is in March, and the 90% confidence rate is 420 m^3^/s (approximately 0.8Q). With WUA_0.8Q_ = 49 × 10^4^ m^2^ as the minimum habitat area required by *Schizothorax* during the spawning period, the WUA under different runoff regulations could meet the demand and the spawning habitat requirements are generally met; the corresponding flow range is from 0.8–2.0 Q. (2) With 60% × WUA_max_ as the minimum habitat area, the WUA spawning habitat area meets 60% × WUA_max_ and above, and the corresponding flow range is from 1.0–1.6 Q. Within this flow range, WUA_(0.3, 0.6__)_ and WUA_(0.6, 0.8__)_ are in a good state. Based on the WUA and CSI distributions, the spawning habitat is in a good state. (3) With 80% × WUA_max_ as the minimum habitat area, the WUA spawning habitat area meets 80% × WUA_max_ and above, and the corresponding flow range is from 1.1–1.4 Q. Within this flow range, WUA_(0.3, 0.6__)_ and WUA_(0.6, 0.8__)_ are in the best state. Based on the WUA and CSI distributions, the spawning habitat is in the best state.

### 3.3. Connectivity Model

We identified the MPS (mean patch size) and NP (number of patches) of the a, b_1_, b_2_, b_3_, and b_4_ classes and calculated the PROX(proximity) of each to analyse the habitat connectivity under different flow conditions.

The connectivity of class a reflects the overall suitable habitat distribution in the region. Figure 13 shows the following: at 0.6–0.8 Q, NP_a_ grows quickly and MPS_a_ is constant, but PROX_a_ maintains a lower value (near 0) because of the scattered patches, resulting in poor connectivity. At 0.8–1.1 Q, NP_a_ and MPS_a_ increase, and the distance between each patch gets closer, PROX_a_ increases slowly, the connectivity enhanced but still poor. At 1.2–1.5 Q, NP_a_ is stable, MPS_a_ experiences a small change, the distance decreases, and the patch distribution is dominated by a large patch cluster, thus PROX_a_ increases significantly and remains at a high state. At this time, the habitat connectivity is great, and concentrated spawning sites might be formed. At 1.6–1.9 Q, NP_a_ remains high but MPS_a_ decreases and the degree of the cluster weakens; thus, PROX_a_ shows an obvious reduction and the connectivity remains in a good state. At 2.0 Q, NP_a_ decreases with a sparse distribution, MPS_a_ is thin, and the degree of patch aggregation significantly declines.

Figure 14 shows the PROX and connectivity of the b_1_, b_2_, b_3_, and b_4_ classes.

B_4_ class: At 0.1–2.0 Q, MPS_b4_ barely changes, and the change in PROX_b4_ is mainly affected by NP_b4_ and the distribution. At 1.3–1.4 Q, with a large NP_b4_ and a small patch distance, PROX_b4_ and connectivity are high. At 1.0 Q, with a lower NP_b4_ but a more concentrated distribution or a large patch cluster, PROX_b4_ is also high.

B_3_ class: At 0.1–2.0 Q, MPS_b3_ barely changes. At 1.3–1.4 Q and 1.6–1.7 Q, NP_b3_ is small with a concentrated distribution, and at 1.2–1.5 Q, NP_b3_ is large with a close distance, all resulting in good PROX_b3_ and connectivity.

B_2_ class: At 0.1–2.0 Q, MPS_b2_ barely changes, and the change in PROX_b2_ is mainly affected by NP_b2_ and the distribution. At 1.2 Q and 1.6 Q, the two peaks of PROX_b2_ appear, at which the patches are clustered and exhibited good connectivity.

B_1_ class: Compared with the former three habitat types, NP_b1_ significantly decreases and is sparsely distributed, resulting in a low PROX_b1_ and poor connectivity.

Considering the connectivity of the five classes comprehensively, the ecological water requirements of *Schizothorax* during spawning were determined to be as follows: (1) At 1.2–1.5 Q, with a high PROX_a_ and a good PROX_b3_ and PROX_b2_, the habitat connectivity is good and the state of the habitat is best. (2) At 1.6–1.9 Q, with a moderate PROX_a_, PROX_b3_, PROX_b2_, and PROX_b1_, the habitat connectivity is moderate and the state of the habitat is good. (1) Using 0.8 Q corresponding to the lower limit of PROX, at 0.8–1.1 Q and 2.0 Q, the habitat connectivity is poor and the state of the habitat is generally satisfactory.

## 4. Discussion

In this study, the habitat simulation method was used to calculate the ecological water requirements in the main stream section of the Yanni wetland in Tibet. The rationality of the habitat suitability curve directly affects the simulation results of the habitat suitability and affects the scientific determination of the ecological discharge. The existing research results for the suitable habitat of *Schizothorax* in Tibet include the following: the spawning area is located in the boundary zone of severe slow current where the eggs are to be laid, the fish eggs need to attach to pebbles, and the water depth range is 0.3–1.5 m. The results of the research confirmed that the spawning ground of the fish should be composed of cobble and rock with sandy land nearby and provided the screening conditions allowing us to establish the hydraulic habitat suitability curve by using the numerical simulation method to ensure the accuracy of the habitat simulation.

In contrast to mountainous, straight river reaches, as the floodplain develops through the research area, scattered spawning grounds and a great variation of the hydraulic conditions along cross-sections arose. Therefore, we selected typical points in the study area to construct the hydraulic suability index. Rapid flow stimulates spawning in *Schizothorax*, slow flow helps young fish forage, and the violent water mixing in the boundary zone between rapid and slow flow is conducive to the fertilisation of fish eggs; therefore, we chose the junction of the main channel and the beach trough as the typical area and established a total of eight typical points along the river.

Using the measured hydrological data, topographic data and water boundary data, the MIKE21 hydraulic dynamic model was established and verified, and used to simulate the dynamic water distribution of typical points from March 1 to April 30 of 2008–2013 in order to calculate the HSI. Based on the hydrodynamic model results, the suitable depth range for *Schizothorax* in Tibet is 0.5–1.5 m (depth = 0.6–1.0 m, HDI ≥0.6), while the suitable velocity range is 0.1–0.9 m/s (velocity = 0.3–0.7 m/s, HVI ≥0.6, which differs from previous estimates of the spawning hydraulic habitat requirements of *Schizothorax* in a southwestern mountainous area (depth: 0.5–1.5 m; velocity: 0.5–2.0 m/s [45]). The main reasons for the difference are as follows: due to the influence of topographic factors, the study area has a wide river channel, flat terrain, developed terraces, and floodplains, and therefore, a slow flow rate. The southwestern mountainous areas are mostly located in canyon areas, with steep slopes, and the cross-section is of the “V” or “U” type. The floodplain does not develop; thus, the flow rate is fast. The establishment of the HDI and HVI enriched the research on the HSI of *Schizothorax* in Tibet, providing a reference for simulating ecological water requirements.

Based on the satellite data, topographic data, field sampling and HCI, the distribution and suitability of cover types in the study area were determined. Additionally, MIKE21 was used to simulate the hydraulic distribution under different runoff conditions to simulate the CSI according to the HDI, HVI, and HCI. Under all regulation scenarios, the water depth is distributed in the main channel, sub-channel, etc., while velocity is distributed in only the main channel, as the sub-channel is mostly in a static state under the influence of the floodplain. Thus, CSI primarily distributes in the main channel. At 0.1–0.5 Q, the water level is within the first step, and the cover type is unsuitable, CSI mainly affected by HCI and is 0; at 0.6–0.8 Q, HCI and HVI increase, while HDI decreases slightly and CSI rises slowly; at 0.9–1.3 Q, HCI maintains the maximum value, HVI increases slowly, HDI is almost unchanged, but the overlapping between the categories is intensified; thus, CSI increases significantly and reaches a maximum at 1.3 Q; at 1.4–2.0 Q, the suitable depth area develops towards the beach, and the overlaps between three suitability categories decrease, causing a reduction in CSI.

As the area of each grid is relatively uniform, WUA is significantly correlated with CSI. Taking the WUA corresponding to the 90% confidence flow rate in the peak spawning period (March) as the lower limit, the appropriate flow range for the runoff regulation is 80–200% × Q (average annual spawning period flow). Due to the wide cross-section, developed floodplain, and changeable terrain, the flow range suitable for spawning is wide when the hydrodynamic conditions are close to the natural state at low flow (0.8–1.0 Q), while at high flow (1.0–2.0 Q), the water level rises slowly as the flow increases and floods the bank on both sides. In contrast to the application of a habitat simulation method to simulate the minimum ecological discharge of rivers [46,47,48], as the runoff regulation is affected by dispatching, meteorology, and other factors, and could be a maximum or minimum flow in extreme cases, we proposed the upper and lower limits of ecological discharge in this study.

Under the influence of topography, the location of scattered spawning grounds changes under different discharge conditions, and their spatial relationship may affect the spawning of *Schizothorax*. Therefore, the connectivity index in the landscape ecology was introduced in this study, and the suitable habitat connectivity under different discharge scenarios was analysed to determine the range of ecological discharge: 80–200% × Q (average annual spawning period flow). The suitable flow range determined by the connectivity and the WUA are the same, but the habitat qualities of the two methods are different at different flow intervals (shown in Table 2).

The WUA and UA connectivity are considered to determine the ecological water demand of *Schizothorax* during the spawning period. According to the differences in the CSI, the suitability of spawning area was divided into four grades, and the habitat status was further divided into three types according to the corresponding WUA and connectivity of each grade—best state, good state, and generally satisfactory state (Table 2)—providing a reference for reasonably determining the dispatching flow.

For the first time, this study proposes to analyse the ecological water requirement of *Schizothorax* from the perspective of the WUA and habitat connectivity, which enriches the habitat simulation factors. Although the results revealed that the WUA and connectivity were consistent in the range of ecological flow, they were inconsistent in the range of flow corresponding to different habitat states. As the spatial distribution of the habitat is very important in the determination of the ecological flow, this method could be extended to calculate the water demand of various fishes. However, when calculating the connectivity, the value of the neighbourhood is key, as it is related to the habitats of fish. It is necessary to further study these conditions for different fish species.

## 5. Conclusions

In this study, we used a hydrodynamic model, habitat model, and connectivity model to calculate the HSI during the spawning of *Schizothorax* in the Yanni wetland and to predict the WUA and connectivity under different runoff regulation scenarios:(1)The model results indicated that the suitable spawning habitat of *Schizothorax* is cobble with nearby sandy land. Additionally, the suitable water depth is 0.5–1.5 m, and the suitable velocity is 0.1–0.9 m/s. The results enrich the research on the HSI of *Schizothorax* in the Brahmaputra River.(2)When the runoff regulation flow was from 424–1060 m^3^/s, the WUA and connectivity satisfied the requirement for spawning under natural conditions with different habitat states.(3)When the runoff was from 424–530 m^3^/s or 848–1060 m^3^/s, the habitat quality generally satisfied the requirements for spawning. When the runoff was from 530–636 m^3^/s or 742–848 m^3^/s, the habitat was in a good state. When the runoff was from 636–742 m^3^/s, the habitat was in a best state.

## Figures and Tables

**Figure 1 ijerph-16-03045-f001:**
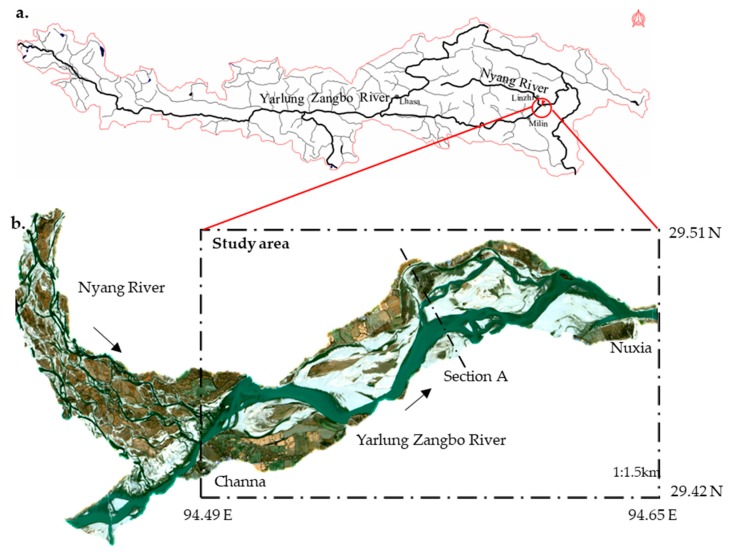
(**a**) Location of the Yanni wetland in the Yarlung Zangbo River. (**b**) The Yanni wetland river network (data from January 2018 Landsat eight satellite images).

**Figure 2 ijerph-16-03045-f002:**
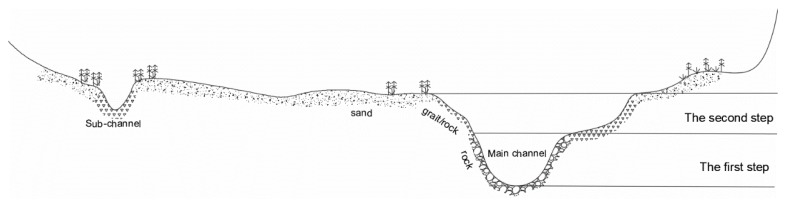
Channel morphology of Section A, which is typical of the study area.

**Figure 3 ijerph-16-03045-f003:**
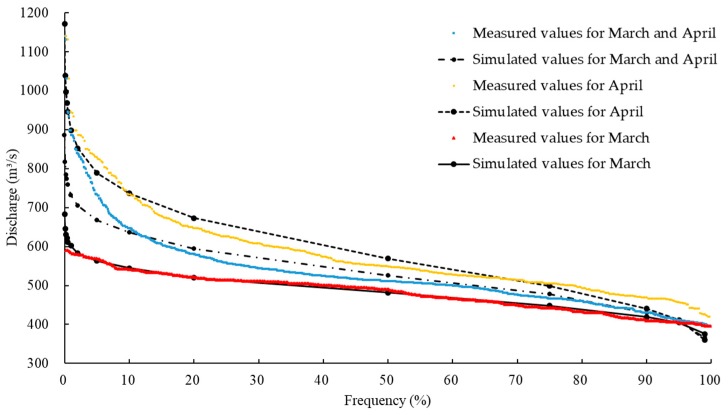
Measured daily discharge during the spawning period from 2000–2013, and the frequency curves simulated by a Pearson type-III curve.

**Figure 4 ijerph-16-03045-f004:**
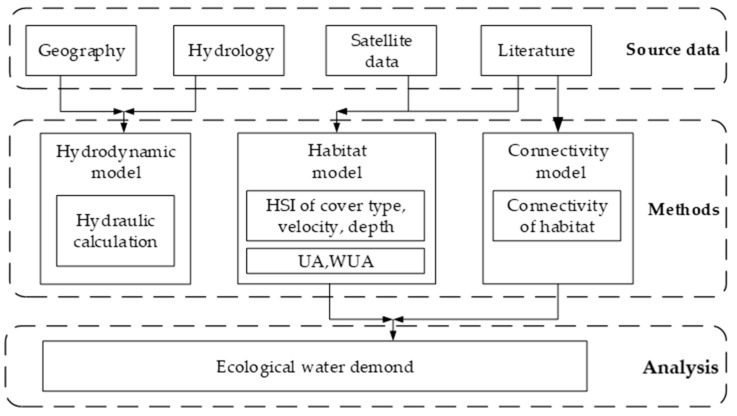
Flowchart for calculating the ecological water demand in the study area.

**Figure 5 ijerph-16-03045-f005:**
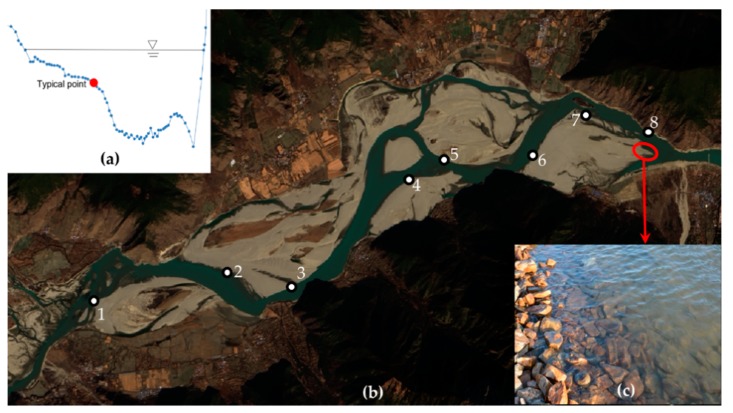
(**a**) shows the extraction position of a typical point in the cross section; (**b**) shows the distribution of typical points along channels, including eight points; (**c**) shows the cover type near the bank.

**Figure 6 ijerph-16-03045-f006:**
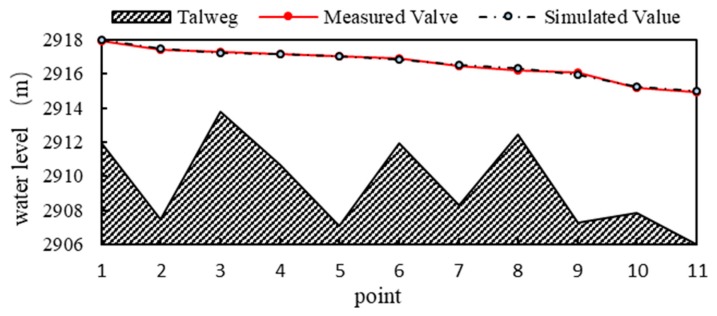
Contrasting the measured water level and simulated value to confirm the model parameter, in which 1–11 are the points shown in Figure 7.

**Figure 7 ijerph-16-03045-f007:**
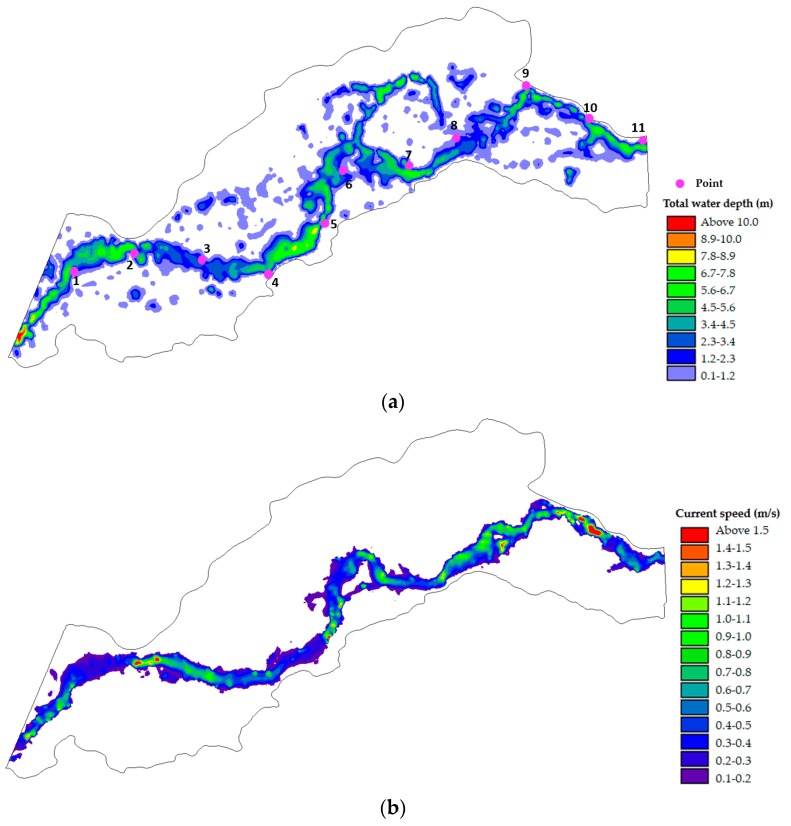
Under natural conditions (1 Q), (**a**) shows the distribution diagram of the water level, and (**b**) shows the flow rate.

**Figure 8 ijerph-16-03045-f008:**
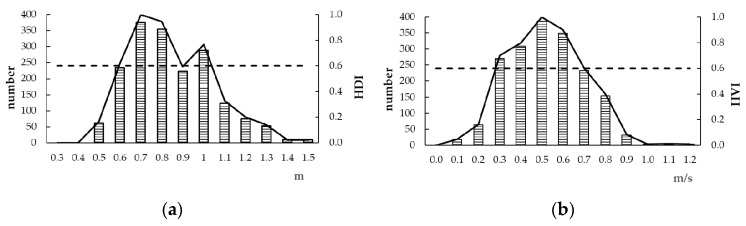
(**a**) shows the sample numbers used for the water depth statistics and the HDI (habitat depth suitability index); (**b**) shows the sample numbers used for the velocity statistics and the HVI (habitat velocity suitability index).

**Figure 9 ijerph-16-03045-f009:**
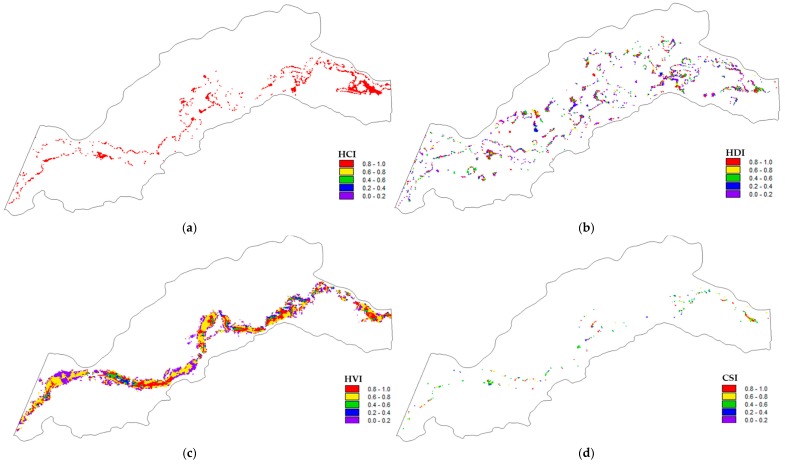
Picture of habitat suitability index distribution under natural conditions including habitat cover type suitability index (**a**); the habitat water depth suitability index (**b**); the habitat velocity suitability index (**c**); and the comprehensive habitat suitability index (**d**).

**Figure 10 ijerph-16-03045-f010:**
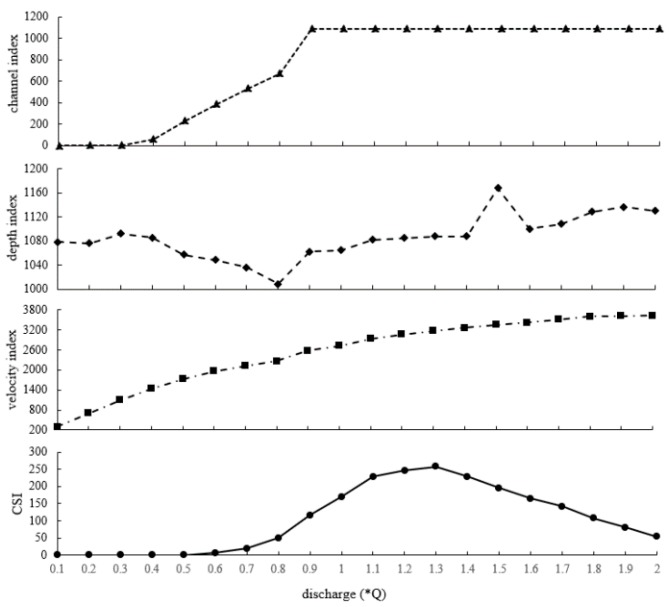
The Variation in HCI_total_ (sum of the habitat cover type suitability index of the study area), HDI_total_ (sum of the depth suitability index of the study area), HVI_total_ (sum of the habitat velocity suitability index of the study area), and CSI_total_ (sum of the habitat comprehensive habitat suitability index of the study area) under different runoff regulation scenarios.

**Figure 11 ijerph-16-03045-f011:**
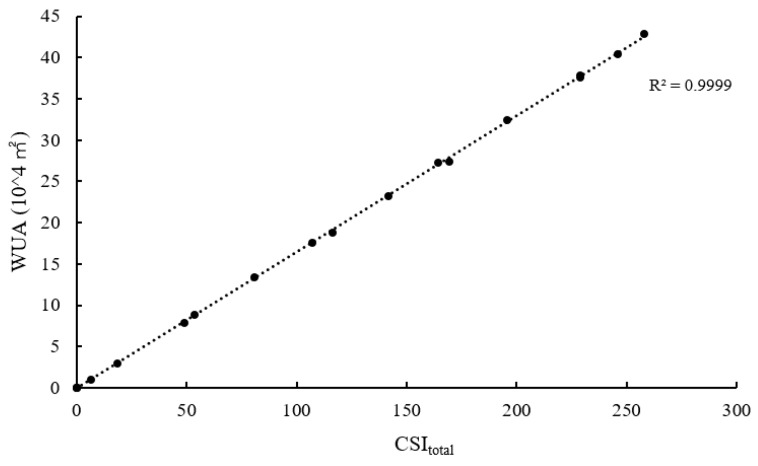
WUA relevance to CSI_total_.

**Figure 12 ijerph-16-03045-f012:**
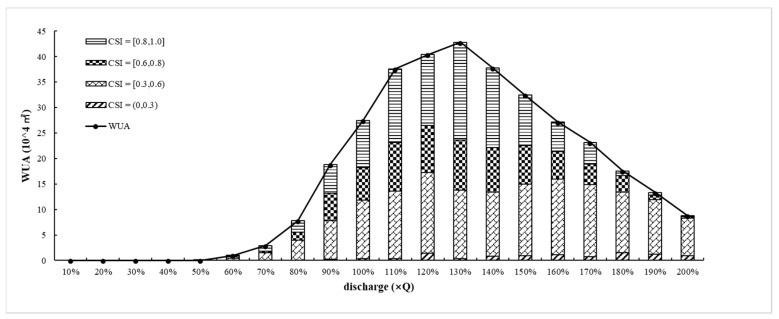
The Variation in the WUA (weighted usable area) under different runoff regulation scenarios.

**Figure 13 ijerph-16-03045-f013:**
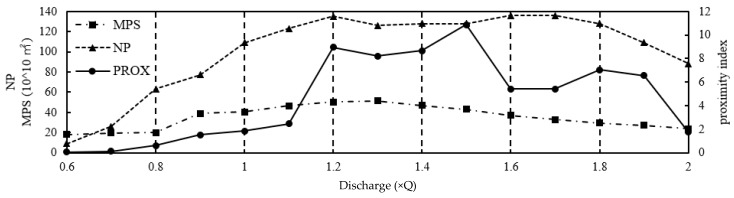
Variation in MPS (mean patch size), NP (number of patches), and PROX(proximity) of class a with runoff regulation.

**Figure 14 ijerph-16-03045-f014:**
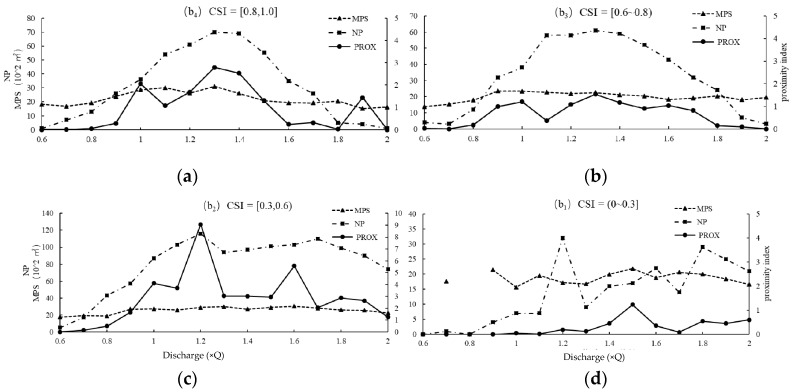
Variation in MPS, NP, and PROX of classes b_4_ (**a**), b_3_ (**b**), b_2_ (**c**), and b_1_ (**d**) with runoff regulation.

**Table 1 ijerph-16-03045-t001:** Categories of suitable area.

Class	CSI ^1^ Range	Suitability Level
a	1.0 > CSI > 0	Total habitat, including all suitable areas
b1	0.3 > CSI > 0	Poor habitat
b2	0.6 > CSI > 0.3	Intermediate habitat
b3	0.8 > CSI > 0.6	Good habitat
b4	1.0 > CSI > 0.8	Best habitat

^1^ CSI—comprehensive habitat suitability index.

**Table 2 ijerph-16-03045-t002:** Ecological water demand of *Schizothorax* during the spawning period.

	×Q ^1^	Habitat State	Corresponding Flow (m^3^/s)
WUA ^2^	0.8–1	Generally satisfactory	530≥; ≥424
	1–1.1	Good state	583>; ≥530
	1.1–1.4	Best state	742>; ≥583
	1.4–1.6	Good state	848>; ≥742
	1.6–2.0	Generally satisfactory	1060>; ≥848
Connectivity	0.8–1.1	Generally satisfactory	583≥; ≥424
	1.1–1.2	Good state	636>; ≥583
	1.2–1.5	Best state	795≥; ≥636
	1.5–1.9	Good state	1007≥; >795
	1.9–2.0	Generally satisfactory	1060≥; >1007
Overall consideration	0.8–1.0	Generally satisfactory	530≥; ≥424
	1.0–1.2	Good state	636>; ≥530
	1.2–1.4	Best state	742>; ≥636
	1.4–1.6	Good state	848>; ≥742
	1.6–2.0	Generally satisfactory	1060≥; ≥848

^1^ The average flow during the spawning period, Q = 530 m^3^/s; ^2^ WUA—weighted usable area.

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
