# Peer review of "The Ecological Water Demand of Schizothorax in Tibet Based on Habitat Area and Connectivity"

_ijerph, 2019, doi:10.3390/ijerph16173045_

Round 1

Reviewer 1 Report

Major comments:

1. The Introduction part should be further improved with a more comprehensive literature review. The contributions of this study should be further highlighted.

2. There are a lot of editing errors throughout the text. I would suggest the authors check the text carefully.

Specific comments:

1. Abstract: the full names of the abbreviations (HIS and WUA) should be given. You cannot expect the readers to know what they stand for before reading the text in the following.

2. The section numbers are not correct.

3. Lines 98, 109: something wrong.

4. Lines 118-119: what does EN stand for? Any reference?

5. Line 176: why not use CHSI for comprehensive habitat suitability index since HSI is used for habitat suitability index?

6. Figure 7: the legend is too small and not clear.

7. Figure 9: for (d), most regions are with the values below 0. Does it mean that the suitability level is worse than poor habitat?

8. Figure 13: the second legend should be NP. The same problem in Figure 14.

9. The conclusion part: the numbers are not correct.

Author Response

Dear reviewer,

We highly appreciate your positive comments and constructive suggestions for the manuscript. Based on your advice, we have revised some relevant content in manuscript.  Please see the attachment

Reviewer 2 Report

This is a very interesting study dealing with the estimation of habitat suitability and the environmental flows during the spawning of Schizothorax in the Yanni wetland (Tibet) using a hydrodynamic model, a habitat model and a connectivity model. The study provides valuable information about the hydrology of an interesting region and presents an interesting approach.

However, the study has two important problems. The first and most important problem is that the methodology is inadequately (very short and confusing) described and therefore it is difficult to make a final judgment about its soundness and its overall merit. The second is the language that should be substantially improved in order to make the study easier to understand.

I have some other comments, which for convenience I included in the commented pdf manuscript file.

Based on the above I suggest a major revision.

(please also see the attached commented pdf file)

Author Response

Dear reviewer,

We highly appreciate your positive comments and constructive suggestions for the manuscript. Based on your advice, we have revised some relevant content in manuscript.  Please see the attachment.

Reviewer 3 Report

1. General Comments: This study investigated the water demand for Schizothorax's activities in a case study in China! The study of important ecological aspects, such as water demand and ecological flow, are important for ecological protection and restoration of natural water regimes (Wang et al., 2018). Wang, H., Wang, H., Hao, Z., Wang, X., Liu, M., Wang, Y. (2018). Water, 10: 326. doi: 10.3390/w10030326 2. Abstract: 2.1. Page 1, Line 12; "This results showed.." Before starting the results, explain the methodology in 1-2 sentences! 2.2. Write keywords alphabetically! 3. Introduction: 3.1. Page 2, Line 74; "The spawning region of Schizothorax in Tibet is is covered..." delete one of "is"!! 3.2. Page 2, Line 88; "No research has focused on the ecological water demand during the spawning period of Schizothorax in Tibet". I think authors tried to bring up the novelty of their study with using this sentence! But please note, you may find a lot of study with same methodology in different regions! Therefore, please remove this sentences, it is unnecessary!   4. Materials and Methods: 4.1. Page 3, Line 96; "Study Area" Please write the geographic coordinates of the wetland 5. Results and Discussion: 5.1. I could not find any strong discussion with this paper!!! Authors just presented their own results without discuss them with previous studies! Compare your results with other studies with same methodology. Try to verify your findings by discuss them! 5.2. I could not find any good explanation about relation between water demand and schizothorax there by this study!! Authors need to more discuss about it.

Author Response

(The authors gave the same response as above.)

Round 2

Reviewer 1 Report

Some of my previous comments are not addressed. For example, 1) the section numbers are still not correct. There are two sections (i.e., Introduction, Materials and Methods) being numbered as "1. ". 2) the references for EN is not added. Moreover, I would suggest the authors reorganize the Results and Discussions parts. More detailed analyses and interpretations should be provided. It is important for a research paper.

Author Response

Dear reviewer,

We highly appreciate your positive comments and constructive suggestions for the manuscript. And are deeply sorry that due to our misunderstanding, some of your previous comments are not addressed accurately. Based on your advice, we have revised some relevant content in manuscript. Our point-by-point response is below.

Point 1: The section numbers are still not correct. There are two sections (i.e., Introduction, Materials and Methods) being numbered as "1. "

Response 1: We've changed the sections’ number.

Point 2: The references for EN is not added.

Response 2: We have added the reference of “EN” in our manuscript in line 124.

Point 3: Moreover, I would suggest the authors reorganize the Results and Discussions parts. More detailed analyses and interpretations should be provided. It is important for a research paper.

Response 3: We have made structural adjustment and language arrangement of various parts of result and discussion to different degrees in line 263 to 520. In which, the mainly modification contents are:

The value basis of HCI is further clarfied in line 290 to 297; the change of PROX was reanalyzed in line 401 to 412; in the discussion section, we add the discussion on the cause of changes of CSI in line 476 to 488.

We thank you for your valuable suggestions and comments that enable us to substantially improve this paper. Once again, thank you very much for your comments and suggestions.

With regards,

Ruidong An

Reviewer 2 Report

The paper seems to be substantially improved. For me it is suitable for publication after only some editorial corrections.

Author Response

Dear reviewer,

Thanks for your reading and valuable suggestions and comments that enable us to substantially improve this paper.

With regards,

Ruidong An

Reviewer 3 Report

It seems the reviewers’ comments have been addressed well.

Author Response

Dear reviewer,

Thanks for your reading and valuable suggestions and comments that enable us to substantially improve this paper. Based on your advice, we have appropriated English editing of our manuscript by AJE (American Journal Experts).

With regards,

Ruidong An